# Evolution of European Resuscitation and End-of-Life Practices from 2015 to 2019: A Survey-Based Comparative Evaluation

**DOI:** 10.3390/jcm11144005

**Published:** 2022-07-11

**Authors:** Spyros D. Mentzelopoulos, Keith Couper, Violetta Raffay, Jana Djakow, Leo Bossaert

**Affiliations:** 1First Department of Intensive Care Medicine, National and Kapodistrian University of Athens Medical School, Evaggelismos General Hospital, 45-47 Ipsilandou Street, 10675 Athens, Greece; 2UK Critical Care Unit, University Hospitals Birmingham, NHS Foundation Trust, Birmingham B15 2TH, UK; k.couper@warwick.ac.uk; 3Warwick Medical School, University of Warwick, Coventry CV4 7AL, UK; 4School of Medicine, European University Cyprus, Nicosia 2404, Cyprus; violetta.raffay@gmail.com; 5Serbian Resuscitation Council, 21102 Novi Sad, Serbia; 6Paediatric Intensive Care Unit, NH Hospital, 26801 Hořovice, Czech Republic; jana.djakow@gmail.com; 7Department of Paediatric Anaesthesiology and Intensive Care Medicine, University Hospital Brno and Faculty of Medicine, Masaryk University, 62500 Brno, Czech Republic; 8University of Antwerp, 2000 Antwerp, Belgium; leo.bossaert@erc.edu

**Keywords:** ethics, resuscitation, terminal care, surveys and questionnaires, emergency care

## Abstract

Background: In concordance with the results of large, observational studies, a 2015 European survey suggested variation in resuscitation/end-of-life practices and emergency care organization across 31 countries. The current survey-based study aimed to comparatively assess the evolution of practices from 2015 to 2019, especially in countries with “low” (i.e., average or lower) 2015 questionnaire domain scores. Methods: The 2015 questionnaire with additional consensus-based questions was used. The 2019 questionnaire covered practices/decisions related to end-of-life care (domain A); determinants of access to resuscitation/post-resuscitation care (domain B); diagnosis of death/organ donation (domain C); and emergency care organization (domain D). Responses from 25 countries were analyzed. Positive or negative responses were graded by 1 or 0, respectively. Domain scores were calculated by summation of practice-specific response grades. Results: Domain A and B scores for 2015 and 2019 were similar. Domain C score decreased by 1 point [95% confidence interval (CI): 1–3; *p* = 0.02]. Domain D score increased by 2.6 points (95% CI: 0.2–5.0; *p* = 0.035); this improvement was driven by countries with “low” 2015 domain D scores. In countries with “low” 2015 domain A scores, domain A score increased by 5.5 points (95% CI: 0.4–10.6; *p* = 0.047). Conclusions: In 2019, improvements in emergency care organization and an increasing frequency of end-of-life practices were observed primarily in countries with previously “low” scores in the corresponding domains of the 2015 questionnaire.

## 1. Introduction

Data from multinational, observational studies suggest a substantial variation in end-of-life practices across European countries, and an increasing frequency of decisions to limit life-sustaining treatments, especially in southern Europe [1,2]. End-of-life practices are evolving continuously as a result of new evidence-based guidelines, publication of randomized controlled trials supporting complex advance care planning (ACP), new laws/policies, and educational activities [3,4,5,6,7,8,9,10,11,12,13,14,15,16,17,18,19,20,21,22,23].

In 2015, we conducted a survey of experts from 31 European countries. We administered a comprehensive questionnaire spanning the following four domains: A: practices/decisions related to end-of-life care; B: access to best available care; C: death diagnosis and organ donation; and D: emergency care organization. Practices and organization of care were scored by numerical summation of positive responses. Results showed substantial variability in country-specific approaches to resuscitation/end-of-life care, indicating the presence of space for evidence-supported progress in all the aforementioned domains [3].

In 2019, we undertook a methodologically improved version of the 2015 survey to address the following questions: (1) How did resuscitation/end-of-life care and emergency care organization evolve over the 2015–2019 period? and (2) Could such evolution, be more marked in countries with “low” (i.e., at or below average) practice/organization scores for 2015?

## 2. Materials and Methods

The current survey conforms with the Checklist for Reporting Results of Internet E-Surveys (https://www.jmir.org/2004/3/e34/; accessed on 20 July 2019 see Appendix A).

Potential study participants from 33 European countries were contacted via e-mail. Participant inclusion criteria comprised nationally and/or internationally recognized, specific, clinical, and/or research expertise in resuscitation and end-of-life care; pertinent evidence should be classifiable in 1 or more of the following categories: (1) European Resuscitation Council (ERC) National Resuscitation Council representative; and/or member of the European Registry of Cardiac Arrest investigators network or other ERC related clinical research networks (e.g., ERC Dispatch Center Survey, Reappropriate Trial, Euro-call); (2) Established researcher in the field: first, second or last author of published, scholarly articles in this field; and (3) At least 3 years of prior service as lead clinician in emergency and/or intensive care.

To reduce the risk of recall/social desirability bias, we aimed for at least three participants from each country. However, this did not constitute an inclusion criterion for country-specific responses. Consequently, responses from countries with just one or two participants were to be anyway included in the data analyses. Following the obtainment of informed consent (see Appendix A), participants were able to electronically access the study questionnaire (Table 1).

Respondents chose either among four options, that is, *never*, *sometimes*, *usually* and *always* or between *no* and *yes* [3]. Subsequently, responses of *never/sometimes* and *usually/always* were categorized as *no* and *yes*, respectively. All data were entered in an original, “anonymized” Excel Masterfile. Original responses were received from 1 September 2019, to 25 October 2019. Participants from each country were asked to reconfirm their answers and provide any missing answers, approximately 3 months after the initial email invitation. Participants were also asked whether they agreed or disagreed with answers provided by other participants from the same country. In cases of disagreement, we encouraged resolution through consensus. The process of data finalization lasted from 1 December 2019 to 31 January 2020. Only consensus-based, country-specific responses were ultimately analyzed, besides the case(s) of having to include responses from just one country-specific respondent. This resulted in a final Excel Masterfile that included a single, country-specific response to each one of the survey questions [3]. For data analysis, we used a dichotomous quantizing approach by grading a positive response with 1 and a negative response with 0 [24].

### 2.1. Questionnaire Structure and Grading

The 2019 questionnaire was organized into four domains (Table 1), precisely like the 2015 questionnaire [3]. Domain A (practices/decisions related to end-of-life care) included subdivisions that included sets of questions pertaining to (1) eight end-of-life practices (e.g., do-not-attempt cardiopulmonary resuscitation (DNACPR), advance directives, advance care planning); (2) end-of life decisions and (3) family presence during resuscitation. Each domain A subdivision could reach a maximum subscore if the responses to all of its subcomponent questions were positive. Domain A score was calculated as the sum of the aforementioned subscores (Table 1).

Domain B, C, and D scores were also calculated by summation of the respective subscores (Table 1). Domain B included subdivisions with sets of questions pertaining to access to (1) best out-of-hospital resuscitation care; (2) best in-hospital resuscitation care; and (3) best postresuscitation care. Domain C subdivisions concerned (1) death diagnosis; and (2) organ donation. Domain D subdivisions included sets of questions related to (1) access to emergency care; (2) defibrillation; (3) organization of out-of-hospital emergency care; 4) organization of in-hospital emergency services; (5) registry reporting of cardiac arrest and (6) education (Table 1).

As further detailed in the footnote of Table 1, Domains A and D of the 2019 questionnaire had a total of 10 sets of questions (concerning specific variables, for example, advance care planning (ACP)) that were not included in the 2015 questionnaire. These “new—2019-only” questions were not taken into account in the calculation of the 2019 Domain A and D scores for the purpose of the below-presented 2019 vs. 2015 comparisons.

### 2.2. Study Outcomes

The primary outcome was the presence/absence of statistically significant differences between 2015 and 2019 in domain A to D scores of all participating countries.

The secondary outcome was the presence/absence of significant differences between 2015 and 2019 domain A to D scores of countries with “low” domain scores in 2015. “Low” 2015 scores were defined as domain scores equal to or lower than the corresponding, overall mean score values of 2015 [3]; more specifically, “low” 2015 scores for domains A, B, C and D were those not exceeding 18, 7, 12 and 23, respectively [3]. Accordingly, “high” (or above average) 2015 scores for domains A, B, C, and D were those exceeding 18, 7, 12 and 23, respectively [3].

The tertiary outcome was the presence/absence of significant differences between changes in “low” 2015 domain scores from 2015 to 2019, and changes in “high” 2015 domain scores from 2015 to 2019.

### 2.3. Additional Data Collection in the Context of Un-Prespecified, Exploratory Analyses

In an effort to determine any potential effect of the coronavirus disease-19 (COVID-19), we asked respondents to determine whether the pandemic could have resulted in changes in any of their original responses to the questionnaire (Table 1). Pertinent data collection started on 15 May 2020 and ended on 29 June 2020.

### 2.4. Protocol Approval and Registration

The study protocol was approved by the Ethics and Scientific Committee of Evaggelismos General Hospital Athens, Greece. The approval was used for study conduct in 32/44 European countries (73%) and Turkey. Countries are listed in the online supplement, along with details for informed consent and personal data protection. The protocol was registered with Clinicaltrials.gov (Identifier: NCT04078815).

### 2.5. Statistical Analyses

The internal consistency of the 2019 and the 2015 questionnaires was assessed by the determination of domain-specific Cronbach’s alpha. The distribution normality of domain scores and subscores was assessed by the Kolmogorov–Smirnov test with Lilliefors significance correction. Data are reported as number, number (percentage) and median (IQR) or mean ± SD unless otherwise specified. Comparisons pertaining to (1) study outcomes; and (2) domain subscores were conducted using an independent samples *t*-test or the Mann–Whitney *U* test.

Bivariate linear regression was used to explore possible associations between (1) the 2019 domain A and D scores with and without the “new—2019-only” questions [3]; and (2) the 2019 variable-specific scores for DNACPR or advance directives and ACP. All analyses were performed using SPSS version 28 (IBM Corporation, Armonk, NY, USA). All reported P values are two-sided. Statistical significance was set at *p* < 0.05.

## 3. Results

### 3.1. Respondents and Countries Participating in the Analysis

A study flow diagram is presented in Figure 1. Initial responses were received from 1 September 2019, to 28 October 2019 from 85 respondents originating from 31/33 European countries (93.9%). The median number (IQR) of respondents per country was 2 (1–4) and ranged from 1 (9 countries) to 9 (1 country). Details on conflicting and/or initially missing responses are presented in Appendix A. Consensus on conflicting responses and provision of initially missing responses was accomplished for 25/33 countries (75.8%), which were ultimately included in the analyses.

### 3.2. Internal Consistency of the 2019 and 2015 Questionnaires

Domains A (end-of-life care practices/decisions), B (access to best resuscitation/postresuscitation care), C (death diagnosis/organ donation) and D (emergency care organization) of the 2019 questionnaire had Cronbach’s alpha values of 0.94, 0.94, 0.63 and 0.74, respectively. Domains A, B, C and D of the 2015 questionnaire, had Cronbach’s alpha values of 0.94, 0.88, 0.61 and 0.78, respectively. Regarding domain C, deletion of a question regarding “organ donation by opting in” in the 2019 questionnaire (Table 1), and deletion of a question about “use of brain death criteria in “out-of-hospital cardiac arrest” in the 2015 questionnaire [3] (Table 1) would result in respective alpha values of 0.70 and 0.68.

### 3.3. Results on Study Outcomes

Results on the primary and secondary outcomes are summarized in Figure 2 and Figure 3, respectively; further details, including scores of variable-specific sets of questions, and additional, subgroup-specific data are presented in Appendix A.

Regarding the primary outcome, domain A and B scores did not differ significantly between 2015 and 2019 (Figure 2). However, domain C score was lower by 1 point in 2019 vs. 2015 (95% confidence interval (CI): 1 to 3; *p* = 0.02); this change was driven by a reduction in the organ donation subscore (Figure 2, Appendix A). In contrast, from 2015 to 2019, domain D score exhibited a significant increase of 2.6 points (95% CI: 0.2 to 5.0; *p* = 0.035) (Figure 2, Appendix A). Regarding domains A and D, the comparable ranges of score values (Figure 2) and coefficients of variation in 2019 (Appendix A) suggest the persistence of the considerable variation in end-of-life practices and emergency care organization observed in 2015 [3].

Regarding the secondary outcome, in countries with “low” 2015 domain scores, domain B and C scores did not differ significantly between 2015 and 2019 (Figure 3). However, from 2015 to 2019, the domain A score increased by 5.5 points (95% CI: 0.4 to 10.6; *p* = 0.047) (Figure 3, Appendix A). The domain D score also increased by 4.7 points (95% CI: 2.1 to 7.3; *p* = 0.009) (Figure 3, Appendix A), thereby driving the “overall increase” reported above and in Figure 2 and Appendix A.

In the context of a “pragmatic”, practice-level presentation, Table 2 presents the main, observed, proportional changes in positive responses to variable-specific questions from 2015 to 2019.

Regarding the tertiary outcome, in countries with “low” 2015 domain A to D scores, all changes in the scores of domains A, C and D from 2015 to 2019 were arithmetically positive and differed significantly from the respective changes determined for countries with “high” 2015 domain A to D scores (*p* ≤ 0.02) (Table 3).

### 3.4. Responses Pertaining Only to the 2019 Survey

Responses to questions included only in the 2019 survey are detailed in the Appendix A. Country-specific, positive response rates of >50% pertained primarily to ACP, shared decision making, dispatcher-assisted CPR and guidance about compressions/ventilation, quality features of prehospital (ambulance) care and educational programs for ethics.

### 3.5. Exploratory Analyses

Linear regression revealed significant associations between 2019 domain A (end-of-life care practices/decisions) and domain D (emergency care organization) scores (adjusted r^2^ = 0.35 to 0.43; *p* ≤ 0.001; Appendix A). There were also strong linear relationships between the 2019 variable-specific score for DNACPR and ACP (r^2^ = 0.68, *p* <0.001; Appendix A) and the 2019 variable-specific score for advance directives and ACP (r^2^ = 0.79, *p* < 0.001; Appendix A). Additional details are reported in the Appendix A.

Results on the effect of COVID-19 were remarkable mainly for changes in access to resuscitation care; for further details see the Appendix A.

## 4. Discussion

The current comparison of responses to the 2015 and 2019 questionnaires from 25 countries revealed no overall significant changes in end-of-life practices and access to best resuscitation/postresuscitation care. There was an apparent decline in organ donation practices in just two countries. There was a significant improvement in the 2019 emergency care organization, driven by countries with “low” 2015 domain D scores. Furthermore, from 2015 to 2019, the frequency of application of end-of-life practices increased significantly in countries with “low” 2015 domain A scores, as opposed to countries with “high” 2015 domain A scores. The considerable variation in practices and emergency care organization noted in 2015 persisted in 2019. As in 2015, a higher 2019 end-of-life practice score was predictive of an improved 2019 emergency care organization [3].

Regarding end-of-life practices, our results are consistent with recent papers on new legislation [25,26] by suggesting a country-level expansion of legal support [3] and application/implementation of DNACPR and advance directives. Integration of DNACPR/advance directives with ACP has been recently advocated in the context of a holistic approach to honoring patient preferences [4,18]. Accordingly, exploratory analyses revealed that the 2019 variable-specific scores for DNACPR and advance directives were predictive of the variable-specific score for ACP.

Current European guidelines support using terminal analgesia and sedation, without hastening death [7,27]. Accordingly, the new Italian law entitled “Rules about informed consent and advance directives” supports prescribing clinically indicated, deep sedation for terminally ill patients, in the presence of valid informed consent [26]. The right to deep continuous sedation is also established by the recent, French Claeys–Leonetti law [28]. The use of sedation and analgesia does not seem to shorten the dying process of terminally ill patients [4,27,29]. Furthermore, terminal sedation and analgesia are recommended by recent Canadian guidelines for the alleviation of any pain/distress after LST withdrawal [30]. This clearly differs from the practice of euthanasia, that is, the intentional and painless termination of the patient’s life upon their request [31]. Despite the fundamental difference as regards the main objective of the intervention (i.e., alleviation of distress vs. termination of life), several authors have expressed concerns about a potential “practice overlap” between deep sedation until death and euthanasia [26,28,32,33,34,35,36,37,38,39,40,41,42]. This could partly explain our results of declining legal support/application of terminal analgesia and sedation, despite the recently reported increase of treatment limitation decisions over time in Europe [1]. Indeed, if certain respondents (subjectively/erratically) viewed “terminal analgesia and sedation” as a form of “euthanasia” [40], they might have provided negative responses for legal support/application [2,31].

In Western countries, out-of-hospital termination-of-resuscitation rules perform well, with proportions of cardiac arrest survivors recommended for termination (i.e., miss rates) of <1% [43,44]. However, miss rates may exceed 6% in countries with lower proportions of in-field defibrillation attempts and shorter in-field resuscitation before patient transportation [44]. Furthermore, the application of termination-of-resuscitation protocols may vary widely at the country level (according to legal support) [6,44], regional healthcare system level (depending on the local frequency of witnessed arrest and bystander CPR) [45], emergency medical service (EMS) or hospital level (according to service-specific or institution-specific resuscitation policies) [44,46] and healthcare professional level (according to pertinent knowledge/expertise, confidence and right/responsibility to decide, possible fear of litigation and personal views) [6,47,48,49]. Such multiple sources of variation and the concurrent inability to issue a “universal/clear-cut” recommendation for a rule [4] may explain our results of the declining application of termination-of-resuscitation protocols.

Organ transplantation prolongs the life of recipients and improves its quality [50]. An ongoing shortage of organs for transplantation has led to the consideration of uncontrolled donation after circulatory death (DCD) [51]. Over the past 16 years, there has been a steady increase in DCD in the United Kingdom, the Netherlands, Belgium and Spain [50,52]. Such country-specific increases in organ donation could not be detected by our survey questions (Table 1).

Automated external defibrillator (AED) availability and use improve survival and neurological outcome after shockable out-of-hospital cardiac arrest [53,54,55,56,57,58]. Barriers and facilitators of bystander defibrillation are related to knowledge/awareness, training, willingness, AED availability/accessibility, medicolegal issues, AED registration and dispatcher assistance [59]. Major problems contributing to AED underutilization comprise AED retrieval distance and time-dependent availability (e.g., functional AED not available at night) [60,61]. Recently proposed improvements included mathematical optimization of AED placement and AED drone delivery to lay rescuers [62,63]. Our results are consistent with an ongoing and expanding effort to increase AED availability in various locations and emergency vehicles and improve AED data collection by creating new AED registries.

Our results of increased defibrillation availability are consistent with the reported improvements in ambulance/pre-hospital level of care. However, the pertinent key determinant was the reported increase in ambulance advanced life support (ALS) (Table 1). Prehospital ALS is cost-effective [64] and efficient paramedic training in ALS interventions may lead to better patient outcomes [65,66,67,68]. Physician-staffed ambulances have been associated with improved neurological outcomes in bystander-witnessed cardiac arrest [69]. EMS physician attendance has been associated with improved survival after cardiac arrest in Norwegian rural areas [70].

As elsewhere detailed [6], registry-based analyses offer valuable insights into regional variation, temporal trends and determinants of cardiac arrest outcomes, the potential efficacy of therapeutic interventions, and the extent of evidence-based clinical practice [70,71,72,73]. The EuReCa projects combined data from the national cardiac arrest registries of 28 European countries and have already reported on key modifiable variables (e.g., bystander CPR and defibrillation rates) affecting patient outcomes [73]. Accordingly, our results suggest an improvement in registry reporting of cardiac arrest from 2015 to 2019.

Responses to domain A and D “2019-only” questions suggested variation in end-of-life practices and emergency care organization, respectively. Variable-specific scores for DNACPR and advance directives were predictive of the variable-specific score for ACP, possibly implying increasing integration of advance directives with ACP [6,18]. Overall and “2019-only” results on emergency care organization indicate a need for multilevel improvement in many countries and are consistent with the observed large variation in cardiac arrest outcomes [71,73].

The current questionnaire was not specifically designed to detect pandemic-induced changes in resuscitation practices and patient outcomes [74,75,76]. Our results are consistent with the recently reported less CPR initiation by bystanders/first aid providers and mobile medical teams, and less AED use [74,75].

### Strengths and Limitations

The current survey’s strengths include coverage of multiple aspects of resuscitation and end-of-life practices/care and comparative determination of their time-dependent changes [3,31].

Limitations (including subjectivity-related bias) exhibit similarities to those acknowledged in our 2015 survey [3]. In 2019, we attempted to increase the number of 1–2 respondents per country in 2015; this was not feasible for 9/25 countries (36%; Appendix A). Consequently, the current work is still primarily based on the opinion of just a few experts from each participating country. We also requested confirmation of original answers and provision of missing answers within 3 months (as opposed to 6 months in 2015). Furthermore, in contrast to 2015, we analyzed only consensus-based data from most (i.e., 16/25; 64%) of the participating countries. Arguably, these methodological differences between the current and the 2015 survey may have limited their comparability. Nevertheless, in 2015, we employed similar criteria for respondent selection and also analyzed consensus-based data from 4/25 (16%) participating countries.

As in 2015, we assumed that respondents had a thorough knowledge of multiple aspects of practice/care or access to information concerning survey items [3]. Furthermore, we employed a dichotomous quantizing approach, which may risk the loss of critical information but also limit respondent subjectivity [3,24]. Arguably, our approach could be regarded as more suitable for questions answerable with a “clear yes or no” (e.g., are DNACPR orders legally allowed?) and less suitable for differentiated questions (e.g., are DNACPR orders applied in the out-of-hospital or in-hospital setting?). However, the determined Cronbach’s alpha values indicated similarly good-to-strong internal consistency of the 2015 and 2019 survey domains [77,78,79]; this further suggests homogenous and replicable patterns of participant responses in both surveys [79], thereby also supporting domain scores’ comparability.

Our results were derived by simple summation of positive responses and should therefore be interpreted with caution as a higher domain A score may not always indicate better practice. Controversial end-of-life practices such as euthanasia and physician-assisted suicide (PHAD) [80,81] should not be considered equivalent to practices aimed at safeguarding patient autonomy, such as advance directives and ACP [4,5,6]. However, euthanasia/PHAD are rarely requested/practiced [2], and pertinent positive responses (increasing domain A scores by ≤15%) were provided by just 4/25 (16%) and 5/25 (20%) countries in 2015 and 2019, respectively.

Finally, in 5/25 countries (20%; respondents, n = 1 and ≥3, in three and two countries, respectively), we noted domain A score changes of >10 points from 2015 to 2019 (Appendix A); pertinent contributing factors could include improved adherence to ethics/end-of-life guidelines published in 2014 and 2015 [5,7], changes in local legislation [25], and recall/social desirability bias [1].

## 5. Conclusions

There was a persisting, substantial variation in resuscitation/end-of-life practices across Europe in 2019, indicating a need for progress, preferably through harmonized legislations, governmental policies, and education/training. There was an overall improvement in emergency care organization from 2015 to 2019, driven by countries with “low” 2015 emergency care organization scores. Significant end-of-life practice changes were also noted in countries with “low” 2015 end-of-life practice scores. As in 2015, higher end-of-life practice scores were associated with better emergency care organization in 2019.

## Figures and Tables

**Figure 1 jcm-11-04005-f001:**
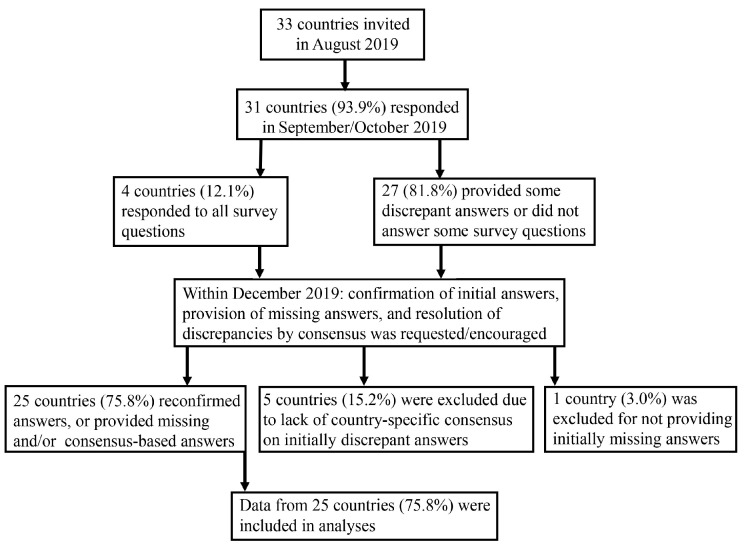
Flow diagram of responses to the 2019 questionnaire.

**Figure 2 jcm-11-04005-f002:**
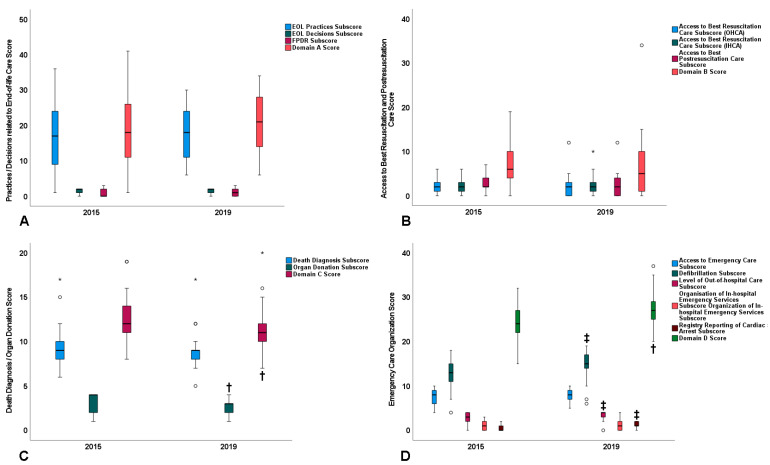
Summary results on the primary study outcome. Boxplot presentation of subscores and scores of domains (**A**–**D**) of the study questionnaire. Data originate from the 25 participating countries. Bars reflect median value; box height reflects interquartile range; bars on top or bottom of the boxes reflect actual range of values; symbols (circles and asterisk) reflect countries with outlier score values, that is, score values outside the range that corresponds to box height plus the bars. †, *p* < 0.05 vs. 2015; ‡, *p* ≤ 0.01 vs. 2015.

**Figure 3 jcm-11-04005-f003:**
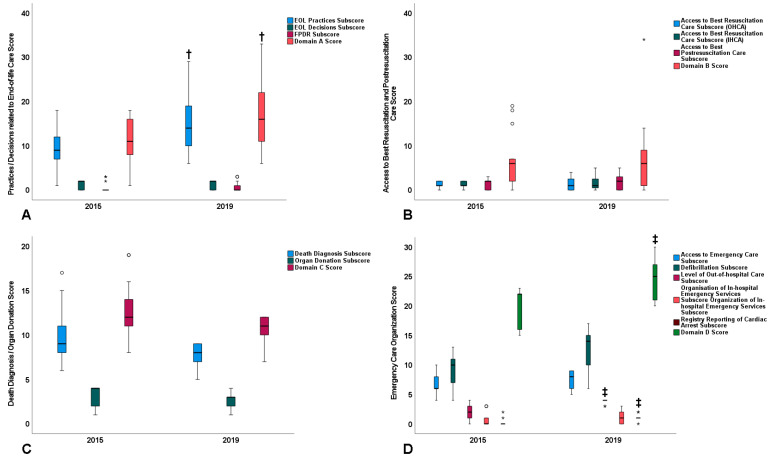
Summary results on the secondary study outcome. Boxplot presentation of subscores and scores of domains (**A**–**D**) of the study questionnaire. Data originate from countries with “low” 2015 domain scores (n = 13 for domains (**A**,**C**); n = 15 for domain (**B**); and n = 9 for domain (**D**)). Bars reflect median value; box height reflects interquartile range; bars on top or bottom of the boxes reflect actual range of values; symbols (circles and asterisk) reflect countries with outlier score values, that is, score values outside the range that corresponds to box height plus the bars. †, *p* < 0.05 vs. 2015; ‡, *p* ≤ 0.01 vs. 2015.

**Table 1 jcm-11-04005-t001:** The 2019 Ethical Practices Questionnaire.

**DOMAIN A. PRACTICES/DECISIONS RELATED TO END-OF-LIFE CARE *** **A1. End-of-life practices** Do-not-attempt cardiopulmonary resuscitation (DNACPR) orders (legally allowed, supported, and applied in your country (3 questions)? applied in single tier, or first and second tier ambulance ^†^ (3 questions)? applied in-hospital? written in medical record? reviewed?) No. of discrete questions (N) = 9; Maximum score if all responses positive (Max. Score) = 9. Advance directives (respect for advance directives legally allowed, and supported in your country (2 questions)? applied in the out-of-hospital, and in-hospital setting (2 questions)?) applied to start/stop cardiopulmonary resuscitation (CPR) in single tier, or first and second tier ambulance (3-questions)? applied to start/stop CPR in-hospital? N = 8, 4-choice; Max. Score = 8. Advance Care Planning (same questions as for advance directives). N = 8, 4-choice; Max. Score = 8. Terminal analgesia/sedation (legally allowed in your country? applied?). N = 2; Max. Score = 2. Termination of Resuscitation protocols (TOR) (legally allowed? applied in single tier, or first and second tier ambulance (3-questions)? applied in-hospital? N = 5, 4-choice; Max. Score = 5. Limitation of in-hospital treatment level (If applied, does it pertain to withholding, and withdrawing CPR (2 questions)? does it include TOR, withholding of invasive treatments, and withdrawing of feeding and hydration?). N = 5; Max. Score = 5. Euthanasia in adults (legally allowed in your country? applied?); euthanasia in children (legally allowed in your country? applied? Physician-assisted suicide (legally allowed in your country?). N = 5; Max. Score = 5. During patient transportation: Is CPR continued in the prospect of organ donation? Is CPR continued in the prospect of access to higher-level treatment (e.g., extracorporeal CPR)? N = 2; Max. Score = 2. Max. Subscore for end-of-life practices (A1) = sum of Max. Scores of 1 to 8 = 44. **A2. End-of-life Decisions** Adults/children: Family participating in Decisions? N = 2; Max. Score = 2. Adults/children: Are end-of-life decisions reached through a shared decision-making process? N = 2; Max. Score = 2. Max. Subscore end-of-life decisions (A2) = sum of Max. Scores of 1 and 2 = 4. **A3. Family presence during CPR** Adults: Family present during CPR? Children: Parents present during CPR? Children: Other family members present during CPR? N = 3; Max. Score = 3. Max. Subscore for family presence during CPR (A3) = 3. **Max. Score for Domain A = sum of max. Subscores of A1, A2 and A3 = 51.** **Questions pertaining to law and those included in A1.6 and A1.7 had 2-choice responses (i.e., *yes* or *no*); Questions pertaining to what is applied had 4-choice responses (i.e., *never*, *sometimes*, *usually*, and *always*).** **B. ACCESS TO BEST RESUSCITATION AND POSTRESUSCITATION CARE**^‡^ **B1–B3. Out-of-hospital (B1) and in-hospital (B2) resuscitation care, and postresuscitation care (B3)** Is access to best available care (including extracorporeal CPR wherever available) affected by age? race? religion? comorbidity? socioeconomic status? urban-rural (area of occurrence)? type of receiving hospital (out-of-hospital setting) or type of treating hospital (in-hospital setting)? minority? language? high-risk presentation (e.g., acute physiology and chronic health evaluation II score > 25 corresponding to >50% mortality probability)? suicide attempt? knowledge of patient’s wish against undergoing CPR? other? The same group of questions was asked about B1, B2, and B3. For each of B1, B2, and B3: N = 13; Max. Score = 13. **Max. Score for Domain B = sum of max. Subscores of B1, B2, and B3 = 39. All questions had 2-choice responses (i.e., *yes* or *no*)**. **C. DIAGNOSIS OF DEATH AND ORGAN DONATION** **C1. Death diagnosis** Legally allowed to diagnose death: physician, out-of-hospital or in-hospital (2 questions)? nurse, out-of-hospital or in-hospital (2 questions)? ambulance person [advanced life support (ALS) provider]? ambulance person [basic life support (BLS) provider]? N = 6; Max. Score = 6. Legally allowed to diagnose death in the absence of obvious signs of death such as rigor mortis or decapitation, and after 20 minutes of asystole without reversible cause: same questions as above; N = 6; Max. Score = 6. Diagnostic criteria for death: Brain death criteria used in out-of-hospital cardiac arrest (after hospital admission) or in-hospital cardiac arrest, and written on death certificate (3 questions)? Cardiorespiratory death criteria used in out-of-hospital or in-hospital cardiac arrest, and written on death certificate (3 questions)? N=6; Max. Score=6. Max. Subscore for death diagnosis (C1) = sum of max. Scores of 1 to 3 = 18. **C2. Organ donation** Heart beating organ donation allowed? Non-heart beating organ donation allowed? Organ donation applied by opting in? Organ donation applied by opting out. N = 4; Max. Score = 4. Max. Subscore for organ donation (C2) = 4. **Max. Score for Domain C = sum of Max. Subscores of C1 and C2 = 22. All questions had 2-choice responses (i.e., *yes* or *no*).** **D. EMERGENCY CARE ORGANIZATION** ^†^ **D1. Access to emergency care** Out-of-hospital: rural areas: emergency number 112 or another (2 questions)? ambulance arrival within 10 min? N = 3; Max. Score = 3. Out-of-hospital: urban areas emergency number 112 or another (2 questions)? ambulance arrival within 10 min? N = 3; Max. Score = 3. In-hospital: emergency number 112 or another (2 questions)? cardiac arrest team arrival within 10 min? N = 2; Max. Score = 2. Max. Subscore for access to emergency care (D1) = sum of Max. Scores of 1 to 3 = 8. **D2. Defibrillation** Legally allowed to defibrillate using an automated external defibrillator (AED): physician? nurse? ambulance personnel? police? On-site responder? Citizen? Other (specify)? N = 7; Max. Score = 7. AEDs available in: single tier ambulance? first tier ambulance? fire cars? police cars? public places? mass gatherings? first responder dispatch projects? other? N = 8; Max. Score=8. Are AED data (electrocardiographic sequence, waveform, time) available in the patient record? N = 1; Max. Score = 1. Ongoing public access defibrillation programs in place? home AED? school AED? in-hospital AED?-Is there a registry of all AEDs, at national or regional level (2 questions)? N = 6; Max. Score = 6. Max. Subscore for defibrillation (D2) = sum of Max. Scores of 1 to 4 = 22. **D3. Organization of out-of-hospital emergency care** Is there a system in place to alert trained lay rescuers (and/or first responders) by text message or app? N = 1; Max. Score = 1. [A] Is dispatcher assisted bystander CPR practiced in rural areas? Does guidance include compressions or ventilations (2 questions)? N = 3; Max. Score = 3. [B] Is dispatcher assisted bystander CPR practiced in urban areas? Does guidance include compressions or ventilations (2 questions)? N = 3; Max. Score = 3. Single tier ambulance: ALS? First tier ambulance: BLS plus defibrillation or ALS (2 questions)? Second tier ambulance: ALS? N = 4; Max. Score = 4. In traumatic cardiac arrest: in single tier ambulance, or first and second tier ambulance: A. Is the intervention unit qualified for prompt confirmation and management of life-threatening injuries (3 questions, one for each type of ambulance)? B. Are specific criteria applied for withholding or terminating resuscitation (3 questions, one for each type of ambulance)? yes-no, specify. N = 6; Max. Score = 6. Max. Subscore for level of out-of-hospital care (D3) = sum of Max. Scores of 1 to 4 = 17. **D4. Organization of in-hospital emergency services** Are in-hospital Rapid Response Teams Programs in place? N = 1; Max. Score = 1 Is CPR feedback, debriefing, and audit applied (3 questions)? N = 3; Max. Score = 3. Is CPR training on the recently dead legally allowed?-is CPR training practiced? N = 2; Max. Score = 2. Max. Subscore for organization of in-hospital emergency services (D4) = sum of Max. Scores of 1 to 3 = 6. **D. EMERGENCY CARE ORGANIZATION** **D5. Registry reporting of cardiac arrest** Out-of-hospital or in-hospital cardiac arrest data reported to a Registry? N = 2; Max. Score = 2. Max. Subscore for registry reporting of cardiac arrest (D5) = 2 **D6. Education** Are there ongoing programs of (a) theoretical education and (b) practice training (e.g., clinical scenario-based) in the field of ethics at pregraduate level (2 questions)? postgraduate level (2 questions)? medical specialty/subspecialty registrar level (2 questions)? specialist level (2 questions)? N = 8; Max. Score = 8. Certified CPR training mandatory for in-hospital healthcare providers: physicians? nurses? other staff? N = 3; Max. Score = 3. Max. Subscore education (D6) = sum of Max. Scores of 1 and 2 = 11. **D7. Research** Enrollment of adults legally allowed without consent in: observational research? interventional research involving drugs? interventional research involving non-drug interventions? N = 3; Max. Score=3. Max. Subscore for research (D7) = 3 **Max. Score for Domain D = sum of Max. Subscores of D1, D2, D3, D4, D5, D6, and D7 = 67. Questions D1.1-3, D2.3, D3.4A, D4.1-3., and D5.1 had 4-choice responses (i.e., *never*, *sometimes*, *usually*, and *always*); all other questions had 2-choice responses (i.e. *yes* or *no*)**

*, Related to the application of the following Ethical Principles: Autonomy, Beneficence, Non-maleficence, Dignity, and Honesty. ^†^, the first tier ambulance corresponds to the capability of BLS plus defibrillation, whereas the second tier ambulance corresponds to the capability of ALS and monitored mechanical ventilatory and hemodynamic support offered by specifically trained and certified personnel. ^‡^, Related to the application of the Principles of Justice and Beneficence. Scores of Domain A subsections A1.3 and A2.2; and scores of Domain subsections D1.3 (question about 112 as emergency number); D2.3; D3.1; D3.2; D3.4, D6, and D7 were not included in the 2019 vs. 2015 comparative analysis, because the corresponding questions were not included in the 2015 Survey [3]. Therefore, for the purpose of this comparative analysis, the Max. Scores for the 2019 domain A and D were 41 and 40, respectively.

**Table 2 jcm-11-04005-t002:** Main proportional (%) changes in positive responses to variable-specific questions from 2015 to 2019.

Domain A—End-of-Life Care Practices/ Decisions	Legally Allowed	Legally Supported	Application out-of-/in-Hospital	Application Related to Start/Stop CPR				Written in Medical Records	Reviewed
				STIER AMB 1st TIER AMB 2nd TIER AMB Hospital					
**DNACPR—all countries (n = 25)**	**16%**	0%	**+28%/+28%**	8%	4%	**24%**	**12%**	**24%**	0%
**DNACPR—low 2015 score (n = 13)**	**31%**	**23%**	**+46%/+46%**	**15%**	**15%**	**31%**	**23%**	**31%**	**15%**
**Ads—all countries (n = 25)**	**20%**	**12%**	**+20%/+24%**	0%	−8%	8%	8%		
**Ads—low 2015 score (n = 13)**	**46%**	**46%**	**0.794872**	8%	**15%**	**23%**	**31%**		
**Term. Analg/Sed - all countries (n = 25)**	−12%		−12%						
**Domain A**	**Legally** **allowed**	**STIER AMB**	**1st TIER AMB**	**2nd TIER AMB**	**Hospital**				
**TOR Protocols—all countries (n = 25)**	−4%	**−12%**	−4%	0%	**−28%**				
**Domain C -Death diagnosis/organ donation**	**Heart beating**	**Non-heart beating**	**Opt in**	**Opt out**	**Opt in and/or Opt out**				
**Organ donation—all countries (n = 25)**	0%	0%	**−48%**	−8%	−8%				
**Domain D -Emergency care organization**	**STIER AMB**	**1st TIER AMB**	**Fire Cars**	**Police Cars**	**Public Places**	**Mass gatherings**	**FRDP**	**Other**	
**Defibrillation Av/ty—all countries (n = 25)**	**32%**	**12%**	**12%**	**20%**	0%	**12%**	−4%	−8%	
**Defibrillation Av/ty—low 2015 score (n = 9)**	**78%**	**33%**	**33%**	**56%**	0%	**22%**	11%	0%	
**Domain D**	**PAD programs**	**Home AED**	**School AED**	**In-hospital AED**	**AED Registry—Nat.**	**AED Registry—Reg.**			
**PAD—all countries (n = 25)**	8%	4%	**28%**	**12%**	**12%**	**36%**			
**Domain D**	**Physician**	**Nurse**	**AMB** **Personnel**	**Police**	**On-site** **Responder**	**Citizen**			
**Legally allowed to defibrillate—low 2015 score (n = 9)**	0%	0%	−11%	0%	−11%	**22%**			
**Domain D**	**STIER—ALS**	**1st TIER—ALS**	**1st TIER—Defibrillation**	**2nd TIER—ALS**					
**AMB level of care—all countries (n = 25)**	**36%**	**36%**	**12%**	0%					
**AMB level of care—low 2015 score (n = 9)**	**67%**	**78%**	**33%**	11%					
	**OHCA**	**IHCA**							
Registry reporting—**all countries (n = 25)**	**36%**	**40%**							
Registry reporting—**low 2015 score (n = 9)**	**33%**	**44%**							

STIER, single tier; AMB, ambulance; DNACPR, do-not-attempt cardiopulmonary resuscitation; ADs, advance directives; Term. Analg/Sed, terminal analgesia/sedation; TOR, termination of resuscitation; Av/ty, availability; FRDP, first responder dispatch project; PAD, public access defibrillation; AED, automated external defibrillator; Nat., national; Reg., regional, ALS, advanced life support; OHCA, out-of-hospital cardiac arrest; IHCA, in-hospital cardiac arrest. Regarding “all participating countries (n = 25)”, proportional changes of ≥12% in ≥3 countries are highlighted in bold script. Domain A: regarding “countries with low 2015 scores (n = 13)” proportional changes of ≥15% in ≥2 countries are highlighted in bold script. Domain D: regarding “countries with low 2015 scores (n = 9)” proportional changes of ≥22% in ≥2 countries are highlighted in bold script.

**Table 3 jcm-11-04005-t003:** Summary results on the tertiary outcome.

	Domain A ΔScore 2015 to 2019	Domain B ΔScore 2015 to 2019	Domain C ΔScore 2015 to 2019	Domain D ΔScore 2015 to 2019
**Low 2015 score countries, n, median (IQR) or mean ± SD**	n = 13 4.0 (−0.5–10.0) *	n = 15 0.8 ± 5.4	n = 13 0.5 ± 1.9 ^†^	n = 9 4.8 ± 4.4 ^‡^
**High 2015 score countries, n median (IQR) or mean ± SD**	n = 12 −2.0 (−6.0–−0.3)	n = 10 −2.8 ± 9.0	n = 12 −3.2 ± 4.3	n = 16 1.4 ± 4.6

Domain A, Practices/decisions related to end-of-life care; Domain B, Access to Best Resuscitation and Postresuscitation Care; Domain C, Death Diagnosis and Organ Donation; Domain D, Emergency Care Organization; ΔScore, Change in Score (from 2015 to 2019). The tertiary outcome comprises the comparison of ΔScores from 2015 to 2019 between countries with low 2015 scores and countries with high 2015 scores; low and high 2015 scores are defined in Methods. *, *p* = 0.01 vs. high-score countries ^†^, *p* = 0.01 vs. high-score countries ^‡^, *p* = 0.02 vs. high score countries.

## Data Availability

“De-identified” datasets used and/or analyzed during the current study are available (in the form of Microsoft Excel Worksheets) from the corresponding author on reasonable request.

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
