# Peer review of "Evolution of European Resuscitation and End-of-Life Practices from 2015 to 2019: A Survey-Based Comparative Evaluation"

_jcm, 2022, doi:10.3390/jcm11144005_

Round 1

Reviewer 1 Report

I am still not convinced by the scientific approach of this study. However, I acknowledge that the authors have seriously responded to the reviewers' comments and have duly taken them into account in revising the manuscript. Since the article presents data on a current topic, I would therefore support the publication of the article in its current version.

Because of the numerous tracked changes, which can be a source of error, I would recommend having the final text proofread by a native speaker.

Author Response

Reviewer 1: I am still not convinced by the scientific approach of this study. However, I acknowledge that the authors have seriously responded to the reviewers' comments and have duly taken them into account in revising the manuscript. Since the article presents data on a current topic, I would therefore support the publication of the article in its current version.

Because of the numerous tracked changes, which can be a source of error, I would recommend having the final text proofread by a native speaker.

Response: We would like to thank the Reviewer for the positive recommendation. Our scientific approach is supported by the cited literature (references 3 and 24 of the revised main paper) and has both strengths and limitations (detailed in page 16 of the revised paper). Finally, we have thoroughly edited the text, while responding to the comments of Reviewer 2.

Reviewer 2 Report

With a great interest i have read the work of Mentzelopoulos et al. on the evolution of European resuscitation and end-of-life practices from 2015 to 2019. As the work I received was in track mode and other formatting changes, I have accepted all changes without of checking it before and I provide pdf document in attachment for lines numbers following.

The authors should be congratulated on a great research idea, identified research gap and great work!

However, my humble opinion is that this work is still missing significant revision of scientific writing and flow in order to be accepted. Furthermore, I would recommend adaptation of text as commented in further text:

Abstract: Please divide abstract into sections (background, methods etc.). Please revise in line 37 “were graded by 1/0 (respectively).” by adding 1 or 0 instead of “/”.

Conclusion in abstract should bring more clear message to reader, without need to check once again what the domain D and A are. Please revise

Introduction:

Please start the introduction with the state of the art in this subject. The first two paragraphs should report on actual situation and what is already known on the subject. The reader is rather surprised with the immediate start of the description of previous work, not being sure what is already known. Please revise.

The aim of the study (line 6-75) should be rather making a question (presenting aim of the study) than suggesting what would be the results. Please revise.

Materials and methods

The information on ethics and data protection (lines 78-87) should be moved to the end of methods, before statistics (after line 189).

Line 89-92 is not clear. Please revise.

Lines 93-98: You are reporting that you aimed to have a minimum of 3 participants from each country, but you included also countries with less than 3 participants. Please revise this paragraph so that it is clear what did you do, and not what do you aimed.

Table 1: “Do-not-attempt cardiopulmonary resuscitation (DNACPR)” – this is usually presented with a shortcut DNR (Do-not-resuscitate). Please consider revision.

What is second tier ambulance? Please revise to more used term.

Terminal analgesia/sedation is rather reported as comfort terminal care or palliative care. Reconsider revision.

“Brain death criteria used in the out-of-hospital”: Please revise. Are there any protocols for brain death diagnosis outside of the hospital? Can we diagnose brain death outside of hospital? (“Determination of brain death is typically made clinically and requires demonstration of the permanent loss of function throughout the brain, including the brainstem, in the absence of factors that may confound the assessment, such as temperature or blood-pressure dysregulation, electrolyte or acid–base disturbances, or toxins or medications. If confounding factors cannot be eliminated, or if the examination cannot be safely or fully performed, ancillary testing is performed (typically in the form of cerebral blood-flow studies that evaluate for the complete loss of cerebral circulation). The determination requires meticulous attention to technique, with careful avoidance of potential pitfalls that may lead to misdiagnosis.”).

“3. In-hospital: emergency number emergency number 112 or another” Please correct.

Table 1 is missing explanation of the shortcuts at the table bottom.

Line 114: “between now and yes” – is it no and yes or now and yes? Please revise.

Results:

The results section is rather missing good flow and scientific writing. Results presented are not easy to follow and to understand. Presentation of results should be made easier and focus on main findings. Division in different domains makes the reading very complex. Tables are complex, and some of them could be exchanged with graphical presentation. The tables footnotes is rather to long and complex.

Figure 1: Please revise word within in the table with more acceptable word (for example in August)

Table 2: P value should always be presented with 3 decimal (0.001). Reporting of >0,99 should be avoided, but 1.000 reported.

Line 274: Please revise, spelling mistake?

Line 351: Please do not start section with “See supplement for additional details”. Please revise.

Statement in the lines from 359-363 should be presented with the table of figure in supplementary.

Line 397: Terminal analgesia is rather supportive that treatment modality.

Line 403: There is no confusion in comfort terminal care. Please revise.

Paragraph line 399-412: Please check the literature once again and revise this paragraph. There is serious difference between comfort terminal care and euthanasia. This has to be clearly stated and explained in order to accept the paper for publication.

Limitations: The main and most important limitation of this work is not reported: This work is based on the opinion of few experts from each country. Please revise by adding this limitation.

Supplementary is well organized.

Author Response

PLEASE SEE THE UPLOADED FILE, WHICH CONTAINS THE CHANGES HIGHLIGHTED IN RED SCRIPT

Reviewer 2: With a great interest i have read the work of Mentzelopoulos et al. on the evolution of European resuscitation and end-of-life practices from 2015 to 2019. As the work I received was in track mode and other formatting changes, I have accepted all changes without of checking it before and I provide pdf document in attachment for lines numbers following.

The authors should be congratulated on a great research idea, identified research gap and great work!

Response: We would like to thank the Reviewer for these positive general comments.

However, my humble opinion is that this work is still missing significant revision of scientific writing and flow in order to be accepted. Furthermore, I would recommend adaptation of text as commented in further text:

Comment #1: Abstract: Please divide abstract into sections (background, methods etc.). Please revise in line 37 “were graded by 1/0 (respectively).” by adding 1 or 0 instead of “/”.

Response: Thank you. We revised the Abstract according to your suggestions. Please note that the revised Abstract has a word count of 221 words. Also, the first sentence of the Abstract’s “Background” section has been modified, in order to become consistent with the first paragraph of the revised Introduction: ”In concordance with the results of large, observational studies, a 2015 European survey…”

Comment #2: Conclusion in abstract should bring more clear message to reader, without need to check once again what the domain D and A are. Please revise

Response: Thank you. The Abstract’s Conclusions now read as follows: "In 2019, improvements in emergency care organization and an increasing frequency of end-of-life practices were observed primarily in countries with previously ″low″ scores in the corresponding domains of the 2015 questionnaire."

Introduction

Comment #3: Please start the introduction with the state of the art in this subject. The first two paragraphs should report on actual situation and what is already known on the subject. The reader is rather surprised with the immediate start of the description of previous work, not being sure what is already known. Please revise.

Response: Thank you. The first paragraph of the Introduction (page 2 of the revised paper) now reads as follows: "Data from multinational, observational studies suggest a substantial variation in end-of-life practices across European countries, and an increasing frequency of decisions to limit life-sustaining treatments, especially in southern Europe [1,2]. End-of-life practices are evolving continuously as a result of new evidence-based guidelines, publication of randomized controlled trials supporting complex advance care planning (ACP), new laws / policies and educational activities [3-23]."

In the second paragraph, we still describe our prior study as it is part of the state of the art and also substantially contributes to the background, rationale and analyses of the current study. Furthermore, to improve this introductory, summary description, we also added the following sentence:

"Practices and organization of care were scored by numerical summation of positive responses."

Comment #4: The aim of the study (line 6-75) should be rather making a question (presenting aim of the study) than suggesting what would be the results. Please revise.

Response: Thank you. The third paragraph of the revised introduction now reads as follows:

"In 2019, we undertook a methodologically improved version of the 2015 survey to address the following questions: 1) How did resuscitation / end-of-life care and emergency care organization evolve over the 2015-2019 period?; and 2) Could such evolution, be more marked in countries with ʺlowʺ (i.e. at or below average) practice / organization scores for 2015?"

Materials and methods

Comment #5: The information on ethics and data protection (lines 78-87) should be moved to the end of methods, before statistics (after line 189).

Response: Thank you. Change performed as suggested - please see page 8 of the revised paper.

Comment #6: Line 89-92 is not clear. Please revise.

Response: Thank you. The second paragraph of the revised Methods (page 2 of the revised paper) now reads as follows:

"Potential study participants from 33 European countries were contacted via e-mail. Participant inclusion criteria comprised nationally and / or internationally recognized, specific, clinical and / or research expertise in resuscitation and end-of-life care;  pertinent evidence should be classifiable in 1 or more of the following categories: 1) European Resuscitation Council (ERC) National Resuscitation Council representative; and / or member of the European Registry of Cardiac Arrest investigators network or other ERC related clinical research networks (e.g. ERC Dispatch Center Survey, Reappropriate Trial, Euro-call); 2) Established researcher in the field: first, second or last author of published, scholarly articles in this field; and 3) At least 3 years of prior service as lead clinician in emergency and / or intensive care."

Please note that this text was actually moved from the supplement to the main paper.

Comment #7: Lines 93-98: You are reporting that you aimed to have a minimum of 3 participants from each country, but you included also countries with less than 3 participants. Please revise this paragraph so that it is clear what did you do, and not what do you aimed.

Response: Thank you. The third paragraph of the revised Methods (page 2 of the revised paper) now reads as follows:

"To reduce the risk of recall / social desirability bias, we aimed for at least three participants from each country. However, this did not constitute an inclusion criterion for country-specific responses.  Consequently, responses from countries with just one or two participants were to be anyway included in the data analyses. Following obtainment of informed consent (see online supplement), participants were able to electronically access the study questionnaire (Table 1). "

Comment #8: Table 1: “Do-not-attempt cardiopulmonary resuscitation (DNACPR)” – this is usually presented with a shortcut DNR (Do-not-resuscitate). Please consider revision.

Response: Thank you. DNACPR is an abbreviation “recommended“ by the European Resuscitation Council. We would therefore prefer to keep it in this paper.

Comment #9: What is second tier ambulance? Please revise to more used term.

Response: Thank you. We have added the following text to the footnote of Table 1: ″†, the first tier ambulance corresponds to the capability of BLS plus defibrillation, whereas the second tier ambulance corresponds to the capability of ALS and monitored mechanical ventilatory and hemodynamic support offered by specifically trained and certified personnel;″

Please note that we used precisely the same term in the 2015 survey (reference 3 of the current, revised paper). Therefore, we would prefer to keep this term as is.

Comment #10: Terminal analgesia/sedation is rather reported as comfort terminal care or palliative care. Reconsider revision.

Response: Thank you. Please note that we used precisely the same term in the 2015 survey (reference 3 of the current, revised paper). Therefore, we would prefer to keep this term as is.

Comment #11: “Brain death criteria used in the out-of-hospital”: Please revise. Are there any protocols for brain death diagnosis outside of the hospital? Can we diagnose brain death outside of hospital? (“Determination of brain death is typically made clinically and requires demonstration of the permanent loss of function throughout the brain, including the brainstem, in the absence of factors that may confound the assessment, such as temperature or blood-pressure dysregulation, electrolyte or acid–base disturbances, or toxins or medications. If confounding factors cannot be eliminated, or if the examination cannot be safely or fully performed, ancillary testing is performed (typically in the form of cerebral blood-flow studies that evaluate for the complete loss of cerebral circulation). The determination requires meticulous attention to technique, with careful avoidance of potential pitfalls that may lead to misdiagnosis.”).

Response: Thank you. Your concern is fully justified. Accordingly, subsection C1.3 of Table 1 now reads as follows: Diagnostic criteria for death: Brain death criteria used ″in out-of-hospital″ or in-hospital cardiac arrest, and written on death certificate (3 questions)? Cardiorespiratory death criteria used in out-of-hospital or in-hospital cardiac arrest, and written on death certificate (3 questions)? N=6; Max. Score=6.

Brain death criteria may be used in out-of-hospital cardiac arrest patients who survive to hospital admission; of course, this pertains to the in-hospital (and not the out-of-hospital) setting; this clarification was actually provided to the respondents during the conduct of the 2019 and the 2015 survey.

Comment #12: “3. In-hospital: emergency number emergency number 112 or another” Please correct.

Response: Thank you. Correction performed.

Comment #13: “Table 1 is missing explanation of the shortcuts at the table bottom.

Response: Thank you. Please note that all the employed abbreviations are defined upon their first use in the text of Table 1; consequently, it is unnecessary to repeat that in the footnote.

Comment #14: Line 114: “between now and yes” – is it no and yes or now and yes? Please revise.

Response: Thank you. Correction performed: “between no and yes” (line 124 of page 7 of the revised paper).

Results

Comment #15: The results section is rather missing good flow and scientific writing. Results presented are not easy to follow and to understand. Presentation of results should be made easier and focus on main findings. Division in different domains makes the reading very complex. Tables are complex, and some of them could be exchanged with graphical presentation. The tables footnotes is rather too long and complex.

Response: In concordance with your comments, we have conducted several major changes in the Results section. More specifically:

A] We have divided the Results section in subsections, which now have the following titles: 1) Respondents and countries participating in the analysis; 2) Internal consistency of the 2019 and 2015 questionnaires; 3) Results on study outcomes; 4) Responses pertaining only to the 2019 survey; 5) Exploratory analyses

Regarding the first subsection: We deleted the last sentence (″In these countries, initially discrepant/missing responses amounted to 33.0±23.1% of a maximum possible number of 179 responses.″), These details are anyway provided in the supplement (Table S1).

Regarding the second subsection: We added the titles of the domains in parentheses to facilitate the reading of the text: ″Domains A (end-of-life care practices / decisions), B (access to best resuscitation / postresuscitation care), C (death diagnosis / organ donation), and D (emergency care organization)…″

Regarding the third subsection: We have replaced Tables 2 and 3 by Figures 2 and 3; further details can still be found in the supplement. We also summarized the previously long text description of ″proportional changes″ in the following (revised) text: ″In the context of a ʺpragmatic,ʺ practice-level presentation, Table 2 presents the main, observed, proportional changes in positive responses to variable-specific questions from 2015 to 2019.ʺ Please note that the proportional changes are still clearly highlighted in Table 2, and there is therefore no real need to repeat this information in the text of the Results. In addition, we deleted the paragraph summarizing the results on the countries with ʺhighʺ 2015 domain A to D scores (ʺResults on countries with ʺhighʺ 2015 domain A to D scores, are presented in supplemental Table S5; domain C total score and subscores were lower in 2019 vs. 2015; there was no difference in domain A, B, and D scores and subscores, besides a higher subscore for registry reporting of cardiac arrest (in domain D) ʺ.) Please note that readers interested in such additional details are still referred to Table S5 from the text of the revised subsection. Lastly, the results on the countries with ʺhighʺ 2015 domain A to D scores were not part of a pre-specified study outcome.

Regarding the fourth subsection: We kept the text from the previous version of the paper.

Regarding the fifth subsection: We revised it according to your subsequent comments

Comment #16: Figure 1: Please revise word within in the table with more acceptable word (for example in August)

Response: Thank you. Correction performed as suggested.

Comment #17: Table 2: P value should always be presented with 3 decimal (0.001). Reporting of >0,99 should be avoided, but 1.000 reported.

Response: Thank you. Table 2 has been deleted. We would like to mention that P>0.99 is considered as entirely appropriate by the Statisticians of the American Medical Association (AMA). The AMA style has been adopted by the journals of the JAMA network.

Comment #18: Line 274: Please revise, spelling mistake?

Response: Thank you. Correction performed as suggested.

Comment #19: Line 351: Please do not start section with “See supplement for additional details”. Please revise. Comment #20: Statement in the lines from 359-363 should be presented with the table of figure in supplementary.

Response (to both Comments): Thank you. We revised the subsection on “Exploratory analyses“ in concordance with your comments. Accordingly, the revised text now reads as follows:

″Linear regression revealed significant associations between 2019 domain A (end-of-life care practices / decisions) and domain D (emergency care organization) scores (adjusted r2=0.35 to 0.43; P≤0.001; supplemental Figures S1A and S1B). There were also strong linear relationships between the 2019 variable-specific score for DNACPR and ACP (r2=0.68, P <0.001; supplemental Figure S1C) and the 2019 variable-specific score for advance directives and ACP (r2=0.79, P <0.001; supplemental Figure S1D). Additional details are reported in the supplement.

Results on the effect of COVID-19 were remarkable mainly for changes in access to resuscitation care according to age/comorbidity, high-risk presentation, delays in response to emergency calls, deactivation of rescuer alert systems  and reduced availability of dispatcher-assisted CPR; for further details see the supplement’s text and Table S6.″

Comment #21: Line 397: Terminal analgesia is rather supportive that treatment modality.

Response: Thank you. We have revised the pertinent text of page 14 to read as follows:

″The use of sedation and analgesia does not seem to shorten the dying process of terminally ill patients [4,28,30]. Furthermore, terminal sedation and analgesia is recommended by recent Canadian guidelines for the alleviation of any pain / distress after LST withdrawal [31].″

Please note that we no longer use refer to terminal sedation and analgesia as treatment modality.

Comment #21: Line 403: There is no confusion in comfort terminal care. Please revise. Comment #22: Paragraph line 399-412: Please check the literature once again and revise this paragraph. There is serious difference between comfort terminal care and euthanasia. This has to be clearly stated and explained in order to accept the paper for publication.

Response (to both Comments): Thank you. We have revised the pertinent text of page 14 to read as follows:

″Furthermore, terminal sedation and analgesia is recommended by recent Canadian guidelines for the alleviation of any pain / distress after LST withdrawal [31]. This clearly differs from the practice of euthanasia, i.e. the intentional and painless termination of the patient’s life upon their request [32]. Despite the fundamental difference as regards the main objective of the intervention (i.e. alleviation of distress vs. termination of life), several authors have expressed concerns about a potential ″practice overlap″ between deep sedation until death and euthanasia [28,29,33-43]. This could partly explain our results of declining legal support / application of terminal analgesia and sedation, despite the recently reported increase of treatment limitation decisions over time in Europe [1]. Indeed, if certain respondents (subjectively / erratically) viewed ″terminal analgesia and sedation″ as a form of ″euthanasia″ [41], they might have provided negative responses for legal support / application [2,32].″

Please note that the pertinent literature (and especially references 33-43) DOES suggest the presence of ″beliefs″ (among healthcare professionals) for a potential ″practice overlap″ between terminal sedation / analgesia and euthanasia. In our opinion, this is incorrect and we have now clearly expressed this in the revised text. However, we still cannot exclude that the presence of such ″beliefs″ has affected our results.

Comment #23: Limitations: The main and most important limitation of this work is not reported: This work is based on the opinion of few experts from each country. Please revise by adding this limitation.

Response: Thank you. We have added the following text to the second paragraph of the subsection ″Strengths and limitations″ (pages 15 and 16 of the revised paper):

″Consequently, the current work is still primarily based on the opinion of just a few experts from each participating country.″

Comment #24: Supplementary is well organized.

Response: Thank you for this positive remark.

Round 2

Reviewer 2 Report

Thank you.

Author Response

RESPONSES TO REVIEWER COMMENTS

We would like to thank the reviewers for their time and effort to improve our paper. Changes performed in the course of the revision are highlighted in red script in the below-provided point-by-point description of our responses to the review comments.  

Reviewer 2: Authors Tried to incorporate most of the comments, however they did not manage to Respond on major comments. Authors recall and are not Willing to adapt the publication, as based on the reference on same subject from 2015 in Resuscitation. My humble opinion that these recommendations are missed in that publication also. Methodological question on out-of-hospital brain death criteria and diagnosis is still opened - brain death criteria can not be made outside of hospital.

If the Editor is of opinion that this work is good enough for publication in Such a good journal as JCM, I would agree with this decision.

Response: We respectfully disagree with the Reviewer, as we think that we have indeed adequately addressed ALL the comments of the preceding review. We would expect that it is understandable that we cannot deviate from any pre-specified (and registered) main methodological feature. For reasons of comparability the methodology of the current survey had to be similar to the methodology of our 2015 survey. Furthermore, we would like to note that our 2015 publication in Resuscitation (reference 3 of the revised main paper) has been well-accepted by the scientific community, as evidenced by its 56 citations in Google Scholar (https://scholar.google.com/citations?view_op=view_citation&hl=en&user=Ndd7a1oAAAAJ&citation_for_view=Ndd7a1oAAAAJ:fPk4N6BV_jEC).

As far as we can understand from the above-provided General Comment, the main expressed concern pertains to the required clarification of the fact that brain death is diagnosed AFTER hospital admission. To clearly indicate this fact, we revised the C.1.3 subsection of Table 1 of the revised main paper to read as follows:

  1. Diagnostic criteria for death: Brain death criteria used in out-of-hospital cardiac arrest (after hospital admission) or in-hospital cardiac arrest, and written on death certificate (3 questions)? Cardiorespiratory death criteria used in out-of-hospital or in-hospital cardiac arrest, and written on death certificate (3 questions)? N=6; Max. Score=6.

We think that it constitutes quite common knowledge that a successfully resuscitated patient with out-of-hospital cardiac arrest can be admitted to the hospital and (more specifically) to an ICU. Following targeted temperature management and hemodynamic stabilization, sedation can be discontinued and if the patient’s neurological status corresponds to a Glasgow Coma Score of ≤4, (a 2-session) brain death testing can be considered.

Finally, we feel somewhat surprised with the Reviewer’s response to our revision, because he/she previously commended us for performing this study. In our opinion, the quality of this survey study is quite sufficient for publication.

We look forward to receiving the Editorial Decision.

Best Regards,

Spyros D. Mentzelopoulos, MD

This manuscript is a resubmission of an earlier submission. The following is a list of the peer review reports and author responses from that submission.

Round 1

Reviewer 1 Report

Dear authors,

the subject dealt with in the manuscript is very current. However, I believe that the presentation of methods and results is confusing and unclear. The responses to the questionnaires are not adequately explained.

In addition, there are numerous typos and repeated words. I also report an error in Figure 1: 27 out of 31 countries are not 12%.
I recommend carrying out a major revision and re-submitting the text.

Reviewer 2 Report

The paper presents the results of a 2019 survey in which various aspects of resuscitation and end-of-life practices in 25 European countries were analysed and compared with the results of a previous survey from 2015. The domains analysed were A: ethical practices, B: access to best available care, C:  death diagnosis and organ donation and D: emergency care organization.

At first glance, that sounds like an exciting project. Reading the paper, however, raised a number of fundamental concerns. Here are some of the most important:

1. The 2015 questionnaire was completed twice within six months by 1-2 experts per country. The 2019 questionnaire was completed by at least 3 experts per country. After 3 months, they were asked to confirm their answers or to add missing data. They also had to assess the answers given by the other country experts. In case of disagreement, the authors encouraged resolution through consensus. Only consensus-based, country-specific responses were ultimately analysed. Compared to the 2015 survey, this seems to be a methodologically different approach that can limit the comparability of the two surveys. But not a word is said about this in the discussion or in the limitations.

2. Referring to the "experts" it is only said that participant inclusion criteria comprised nationally and/or internationally recognised, specific, clinical and/or research expertise in resuscitation and end-of-life care. For further details, reference ist made to the supplement, which I, as the reviewer, did not have. This is not sufficient information.

3. Positive/negative responses were graded by 1/0 (respectively). Domain subscores/total scores were calculated by summation of practice-specific response grades. Such a dichotomous approach may be appropriate for questions that can be answered with a clear yes or no (e.g.: Are DNACPR-orders and/or advance directives legally recognized?). However, differentiated questions (e.g.: Are DNACPR-orders and/or advance directives actually applied in the out-of-hospital / in-hospital setting?) cannot be evaluated with a simple yes/no answer. The questions are also very different: Whether there is a law for advance directives can be answered objectively. Whether this actually supports advance directives is already a subjective assessment (e.g.: I know legal regulations that place such high demands on a binding living will that I would see them as more of a hindrance when it comes to respecting the patient's will). And the question of whether advance directives are applied in practice requires a differentiated empirical analysis and cannot be answered with a simple yes/no answer for an entire country (probably not even for a single institution). To make statements e.g. about the “ethical practices” by simply adding up the yes answers to these various questions doesn't seem scientifically appropriate to me.

4. The underlying idea of ​​the study is that more yes answers indicate better practice. In the introduction, for example, a “need for improvement” is mentioned in connection with countries with “low” 2015 scores. This requires a clear understanding of what constitutes good practice. This is not addressed in the paper. Looking at the individual questions, there are some where it may be relatively uncontroversial that a positive answer indicates good practice (e.g. that DNACPR orders are documented in patient records and regularly re-evaluated). For other questions, however, this is likely to be highly controversial (e.g. whether euthanasia and assisted suicide are legally permitted). Since only the yes answers are added together (e.g. for the evaluation of “ethical practices”), this means that a country could theoretically “make up for” deficits in the documentation of DNACPR orders through legal regulation of euthanasia and assisted suicide. That seems pretty absurd to me.

5. In the discussion, the terms “terminal analgesia/sedation” and “palliative sedation/analgesia” are used synonymously. It is not taken into account that the terminology can go hand in hand with different understandings and that there is also a broad ethical debate on this. Both terms are also equated with a deep sedation until death. In particular, the term "palliative sedation" can also include intermittent or mild sedation.

Conclusion: I am not convinced by the scientific approach of this study. In particular, I have the impression that the authors are relatively blind to the ethical implications of their study. Without major revisions (particularly in view of concerns 3, 4 and 5), are needed.